# Ecological Interactions Between *Camellia oleifera* and Insect Pollinators Across Heterogeneous Habitats

**DOI:** 10.3390/insects16030282

**Published:** 2025-03-08

**Authors:** Linqing Cao, Qiuping Zhong, Chao Yan, Xiaoning Ge, Feng Tian, Yaqi Yuan, Jinfeng Wang, Jia Wang, Shengtian Chen, Hong Yang

**Affiliations:** 1Experimental Center of Subtropical Forestry, Chinese Academy of Forestry, Xinyu 338000, China; caolq1991@126.com (L.C.);; 2Key Laboratory of Cultivation and Utilization for Oil-camellia Resources, Xinyu 338000, China; 3Institute of Forest Resource Information Techniques, Chinese Academy of Forestry, Beijing 100091, China

**Keywords:** *Camellia oleifera*, flower-visiting insects, pollination biology, adaptation

## Abstract

*Camellia oleifera* is an important woody oil plant in southern China. However, owing to a sharp decline in the number of wild pollinators, there are great limitations to the pollination and fruiting of *C. oleifera*. Therefore, it is important to evaluate the adaptations of insect pollinators to the pollination biological characteristics of *C. oleifera*. We investigated the flowering process, floral characteristics, and other biological traits of *C. oleifera* and analysed the differences in the types and quantities of flower-visiting insects in different habitats. By combining the characteristics of insect population and pollination behaviour, dominant pollinators varied by habitat, with *Apis cerana* and *Phytomia zonata* being the most frequent. Stamen length was an important factor for flower-visiting insects, and recommendations were presented for the protection of *C. oleifera* pollinators.

## 1. Introduction

Insects play an important role in the reproduction of cross-pollinated plants, with pollinator species diversity serving as a critical determinant of pollination service efficacy [1]. Both the taxonomic richness and abundance of flower-visiting insects directly correlate with fruit set outcomes in plant populations [2,3,4]. Globally, declines in pollinator populations threaten crop yields by disrupting mutualistic pollination networks–complex, nested systems that stabilize plant-pollinator interactions [5]. Co-evolutionary dynamics between plants and pollinators drive reciprocal adaptations: plants modulate floral traits (e.g., morphology, pigmentation, scent profiles, and nectar chemistry) to optimize pollinator attraction, while insects evolve behaviours that enhance pollen transfer efficiency [6]. For instance, floral displays with larger corollas or high-contrast pigmentation attract greater pollinator diversity and visitation rates [7,8], whereas scent preferences vary taxonomically. Hymenoptera (e.g., bees) favor aromatic volatiles, whereas Diptera (e.g., flies) respond to sulfuraceous compounds [9,10]. Nectar composition further shapes pollinator selectivity, with differential responses to sugar concentration and oligosaccharide profiles across insect taxa [11,12]. The interaction between floral characteristics and pollinating insect behaviour promotes adaptative convergence.

*Camellia oleifera* (Theaceae), a perennial evergreen shrub widely cultivated in China, is a keystone woody oil crop prized for its seed-derived unsaturated fatty acids [13,14]. As an obligate outcrossing species, its reproductive success hinges on insect-mediated pollination, wherein pollen deposition dynamics (quantity, quality, and temporal precision) are tightly linked to pollinator activity patterns [15]. The species flowers from October to December, a period coinciding with reduced insect activity in temperate regions. Primary pollinators include Hymenoptera (Apidae and Vespidae) and Syrphid flies [16], with Apidae bees historically regarded as optimal pollinators. Despite its ecological and agricultural value, *C. oleifera* faces severe pollination bottlenecks. Its floral rewards (nectar and pollen) contain high levels of oligosaccharides (e.g., raffinose, stachyose, and manninotriose), which induce larval rot and adult dyspepsia in domesticated bees (*Apis mellifera*), leading to apiarist avoidance of *C. oleifera* plantations [17,18]. Consequently, wild pollinators (e.g., *Andrena* spp. and *Xylocopa* spp.) have been proposed as alternative agents [19]. However, intensive agricultural practices—including monoculture tillage, herbicide application, and synthetic fertilizer use—have decimated wild pollinator populations, exacerbating pollen limitation in *C. oleifera* orchards [20]. This pollinator scarcity poses a critical constraint on oil yield sustainability. Therefore, studying the pollination ecology of *C. oleifera* is important as it can reveal which groups of visiting insects can potentially provide good pollination services to *C. oleifera* varieties, considering their floral attributes, reducing pollen limitations, and increasing fruit set. This is an effective strategy to address the current bottlenecks in the development of the *C. oleifera* industry. We investigated the morphology and floral biology of the flower-visiting insects of some varieties of *C. oleifera* to identify the high-potential pollinators based on the floral attributes of the studied varieties and the behaviour and foraging of the insects on the flowers. Our results will support management programs for the conservation of potential pollinators of *C. oleifera*, with a consequent increase in fruit production.

## 2. Materials and Methods

### 2.1. Study Area

Experimental sites comprised three plots. Plot 1 is located in Youcheng Township, Poyang County, Shangrao City (116°55′4″ E, 29°10′18″ N). The base has an area of 340 ha, and the soil type is yellow soil. The base was established in 2012 with two-year-old *C. oleifera* seedlings. Management of the *C. oleifera* base involved ecologically digging trenches to apply organic fertiliser, which was performed annually after fruit harvesting. The base was maintained with one weeding and fertilising session in the fall (October) and a pruning session in winter (December). A rural road crossed the base. Plot 2 is situated in Qianshan Town, Fenyi County, Xinyu City (114°39′5″ E, 27°44′52″ N). The base has an area of 22 ha, and the soil type is red-yellow soil. This base was planted with three-year-old container-grown *C. oleifera* saplings in 2017. Management included two weeding and fertilising sessions in May and September each year, with the application of organic fertiliser supplemented by compound fertiliser in the trenches and one pruning session in the winter. The base was bordered by a rural road to the north. Plot 3 is located in Fengtian Town, Anfu County, Ji’an City (114°42′58″ E, 27°23′34″ N). The base has an area of 30 ha, and the soil type is red soil. The base was established in 2012 with two-year-old *C. oleifera* seedlings. Management practices included the application of compound fertiliser in the trenches during winter, one weeding and fertilising session in fall, and one pruning session in winter. The base was bordered by a provincial road to the east. The base was bordered by a provincial road to the east. All three experimental plots contained the Changlin series of the improved *C. oleifera* variety, and other meteorological data are shown in Table 1.

### 2.2. Experimental Methods

#### 2.2.1. Floral Morphology and Biology

Our field survey was conducted from 5 to 15 November 2024 at plot 3. Twenty open flowers per variety were randomly selected, and the following traits were measured using a calliper (precision: 0.01 cm): corolla diameter, androecium diameter, stamen length, pistil length, and style length.

Pollen viability was measured by the iodine-potassium iodide staining method [6]. The procedure is as follows: 20 budding flowers of different *C. oleifera* varieties were selected for hydroponics, and anthers were collected from the first day of flowering and then every 24 h. Using a pipette (Thermo Fisher Scientific, Waltham, MA, USA), add a drop of I_2_-KI solution onto a slide. Gently shake the anthers with tweezers (Thermo Fisher Scientific, Waltham, MA, USA) to evenly disperse the pollen into the I_2_-KI solution on the slide. Cover the slide and observe the slide under an optical microscope (Yi Yuan Optical Instrument, Shanghai, China). For each slide, observe 5 fields of view, ensuring each field contains at least 20 pollen grains. Perform three replicates. If the pollen stains blue, it indicates viability. Count the number of blue-stained pollen grains and calculate pollen viability using the formula: Pollen viability (%) = (Number of blue-stained pollen grains/Total number of pollen grains in the field) × 100.

Stigma receptivity was evaluated using the benzidine-hydrogen peroxide method [6]. The procedure is as follows: 20 budding flowers of different *C. oleifera* varieties were selected for hydroponics, and stigmas were collected from the first day of flowering and, then every 24 h, immersed in a concave slide containing a benzidine-hydrogen peroxide solution (composed of 1% benzidine, 3% hydrogen peroxide, and water in a 4:11:22 ratio). Under a stereo microscope (Motic, Xiamen, Fujian, China), the number of bubbles and the colour change of the stigma were observed. A stigma that quickly produced bubbles and turned blue immediately indicated strong receptivity. Moderate receptivity was indicated by fewer bubbles and a slower color change. If there were no significant change in the stigma before and after immersion, it indicated no receptivity.

Nectar total sugar content was measured using the phenol-sulfuric acid colourimetric method. HPLC method for the determination of monosaccharide components in Nectar, Thermo ICS 5000+ ion chromatography (Thermo Fisher Scientific, Waltham, MA, USA) with an electrochemical detector was employed for the analysis in nectar samples [17]. The analyses included sugars such as sucrose, fructose, glucose, galactose, raffinose, stachyose, and manninotriose. 400 μL of nectar sample (three biological replicates of each sample).

#### 2.2.2. Floral Visitor and Potential Pollinator

##### Survey

The survey was conducted from 5 to 15 November 2024. On sunny, windless days, 10 *C. oleifera* trees were selected from each study plot. Survey methods combined camera photography, net capture, and trapping devices to investigate insect species and numbers. The 10 focal trees at each site remained the same throughout the experiment for trapping devices. However, camera photography and netting surveys were conducted on adjacent trees (within 10 m of focal trees) to avoid disturbing the focal observations. From 8:00 AM to 6:00 PM, flower-visiting insects on *C. oleifera* were observed, photographed, and recorded. Insects were captured using insect nets and killed with ethyl acetate in bottles. The trapping device method (Figure 1) was based on and improved upon Li’s method [19] by adding some solutions that attract insects (Surfactant 12 mL, white sugar 100 g, phenol 12 mL, water 1 L). A homemade insect-trapping device was placed 50 cm from the *C. oleifera* tree, featuring four yellow-trapping plates arranged vertically in the east, west, south, and north directions at heights of 0.5, 1.0, 1.5, and 2.0 m above the ground. Each trapping plate was filled with a homemade solution (approximately 1/3–1/2 of the volume of the plate). After 24 h, insects trapped on the plates were collected. The collected insects were transported to the laboratory, prepared as pinned specimens, and identified and counted based on relevant literature.

Hill numbers help capture the contribution of rare species to overall diversity. We calculated beta diversity based on four of Hill’s numbers [21,22,23]:
Species richness (*q* = 0): *N*_0_ = *S*Exponential Shannon index (*q* = 1): *N*_1_ = *exp* (−$\sum_{i=1}^{s}P_{i}\ln\;P_{i}$)Inverse Simpson index (*q* = 2): *N*_2_ = 1/$\sum_{i=1}^{s}{\left(P_{i}\right)}^{2}$Inverse Berger–Parker index (*q* = 3): *N*_3_ = *N*/*N_max_*
where $P_{i}$ represents the proportion of individuals of the *i*th species out of the total number of individuals, *S* is the total number of species, *N* is the total number of individuals across all species, and *N_max_* is the number of individuals of the most abundant species.

As indicated by Chao [23], *N*_0_ takes into account the number of species in the assemblage but not their relative abundances; *N*_1_ weights species in proportion to their frequency of occurrence and can be interpreted as the number of “common species” in the assemblage; *N*_2_ is weighted toward the most abundant species and represents the number of very abundant species in the assemblage; *N*_3_ can be interpreted as the effective number of the most dominant species in the community.

##### Visit Behaviour and Body Pollen Load

To sample insects on flowers, three trees of each variety were selected. Observations were conducted with one person positioned in front and another behind each tree. The observations were carried out during the peak activity period of pollinators, from 10:00 AM to 3:00 PM, to ensure consistency and capture the highest insect activity levels. Each tree was observed for 10 min, and observations were conducted continuously over 3 days, during which the species and number of insects per unit time were counted.

Single-flower visit duration refers to the time each insect spends on a single flower. The timing starts when the insect lands on the flower and stops when it leaves. During the observation period, if an insect stayed on the flower for less than 1 s, it was uniformly recorded as 1 s. A total of 50 insects were recorded during this process.

The flower-visiting frequency was defined as the number of flowers visited by each insect within 1 h. The timing started when the insect landed on a flower, and the number of flowers visited within 5 min was recorded using a handheld counter, with the measurements repeated 10 times. For each floral visitor, we calculated the flower-visiting frequency (VF) as follows: VF = number of individuals × number of flowers visited per individual/total observation time [24].

The pollen load of floral visitors was randomly collected, and 2–3 insects were immersed in a centrifuge tube containing 5 mL of ethanol solution. The samples were then subjected to ultrasonic shaking for approximately 5 h to dislodge pollen grains from the pollinators’ body surfaces, resulting in a pollen suspension. A 10 µL aliquot of the suspension was precisely pipetted into a hemocytometer and counted under a microscope for 10 replicates.

For comparative analysis of the body characteristics of the flower visitors, vernier callipers (VGS Technology, Shanghai, China) were used to measure the body length and width of the collected insects, with the measurements repeated 10 times for each species.

### 2.3. Statistical Analysis

Data analysis and plotting were performed using SPSS version 26.0, Origin 2021, and R4.4.0. One-way analysis of variance (ANOVA) was used to compare the floral traits of different *C. oleifera* varieties, including corolla diameter, androecium diameter, stamen length, pistil length, style length, total sugar content, and oligosaccharide content, followed by Duncan’s multiple comparison test. One-way ANOVA was also used to compare the exponential Shannon index, inverse Simpson index, and inverse Berger–Parker index among different plots, with Duncan’s method applied for multiple comparisons. Additionally, one-way ANOVA was employed to compare high-potential pollinator traits, including body length, body width, pollen load, flower visitation frequency, and average single-flower visit duration, with Duncan’s method used for multiple comparisons. In all statistical analyses, a level of *p* < 0.05 was considered statistically significant.

Redundancy analysis (RDA) was used to examine the relationship between *C. oleifera* floral traits (explanatory variables) and insect population characteristics (response variables). Floral traits included 14 variables, such as corolla diameter, androecium diameter, stamen length, pistil length, style length, sucrose, fructose, glucose, galactose, raffinose, stachyose, manninotriose, total sugar content, and oligosaccharide content. Insect population characteristics included three variables, such as DN (number of Diptera insects), HN (number of Hymenopteran insects), and IPN (number of insect population). Data processing was performed using R 4.4.0, with the vegan package used for RDA and the adespatial package used to calculate the contribution of each variable. The RDA method can directly reveal the linear relationship between the explanatory variable and the response variable and reduce the dimension of the fitting value matrix through PCA, making the results more intuitive and easier to interpret.

## 3. Results

### 3.1. Biological Characteristics of C. oleifera During the Flowering Period

The floral lifespan of individual *C. oleifera* flowers typically lasted 5–8 d. The patterns of single-flower blooming dynamics were essentially identical across the six varieties. On clear days, the petals gradually opened from approximately 9:00 to 10 AM and were almost fully open by 12:00 PM. The anthers began to release pollen 2–3 h after the flowers opened. On the first day of flowering, the anthers appeared light yellow, and only a negligible amount of nectar was observed around the ovary. By the second day, the anthers turned from light yellow to golden yellow, accompanied by a gradual increase in nectar production. By the third day, the petals spread outwards, the anthers were fully mature, and a large amount of nectar appeared inside the flower, ranging from 100 to 400 μL. On the fourth day, brown spots appeared on the petals, which began to wilt slightly. The anthers turned brown, and the nectar turned sour or dropped onto the petals. By the fifth day, the petals began to fall, the filaments withered progressively from the outer to the inner parts, and black spots appeared on the stigma. Between the sixth and eighth days, all the petals had completely withered, and the style gradually dried and contracted from top to bottom.

Pollen viability and stigma receptivity directly influence the pollination, fertilisation, and fruiting of *C. oleifera*. From the first day of flowering, both pollen viability and stigma receptivity were high (Figure 2, Table 2). Over time, the pollen viability of *C. oleifera* gradually decreased, retaining some vitality until the seventh day. Stigma receptivity also declined gradually from the fourth day and was completely lost by the seventh day of flowering.

The flowers of *C. oleifera* are white, with corolla diameters ranging from 53 to 67 mm, androecium diameters of 14–21 mm, stamen lengths of 14–19 mm, style lengths of 7–11 mm, and pistil lengths of 11–15 mm (Figure 3 and Figure 4). There were varying degrees of difference in floral traits among the six *C. oleifera* varieties (Figure 4). For example, Changlin 4 and Changlin 3 have larger corolla diameters, androecium diameters, and stamen lengths, whereas Changlin 40 has a longer style and pistil length. Changlin 53 and Changlin C31 had smaller corolla diameters, androecium diameters, style length, and pistil length.

The sugar composition of *C. oleifera* nectar is shown in Figure 5. There were some differences in the sugar composition ratios among the different *C. oleifera* varieties. Overall, the sugar components of the nectar were as follows: sucrose > raffinose > stachyose > glucose > fructose > manninotriose > galactose. The total sugar content in *C. oleifera* nectar ranged from 250 to 382 mg/g, with Changlin 53 having the highest content and Changlin 3 and Changlin C43 having lower levels. The other varieties had a total sugar content concentrated at approximately 340 mg/g. The oligosaccharides in *C. oleifera* nectar that can cause bee poisoning (raffinose, stachyose, and manninotriose) make up 30–45% of the total sugars, with Changlin C31 having the highest content and Changlin 3 having the lowest.

### 3.2. Insect Survey in C. oleifera Populations: Composition and Diversity

A total of 22 species of flower-visiting insects, representing 2 orders, 8 families, and 14 genera, were recorded across the 3 plots (Table 3). Overall, the abundance of Diptera (62.54%) was higher than that of Hymenoptera (37.46%). These insect species visited *C. oleifera* flowers belonging to 3 groups—bees (2 spp.), wasps (9 spp.), and flies (11 spp.)—that collected pollen and/or nectar (except for some of the flies). Common groups found in all three plots were Apidae, Vespidae, Syrphidae, Calliphoridae, Sarcophagidae, and Muscidae.

However, the composition of insect species varies across different plots. In plot 1, flies constitute the predominant species, followed by wasps and bees. *Phytomia zonata* was the richest species in plot 1. In plots 2 and 3, the insect composition was similar, with flies being the most numerous, followed by bees and wasps, particularly *Apis cerana* and *P. zonata*, which were abundant species.

We used the Hill numbers to evaluate the biodiversity information of the three plots. The Hill numbers are very rich in information since they combine information on species richness, species rarity (species relative abundances), and species dominance. The Hill numbers allow a complete characterization of the diversity of a community. The characteristics of major flower-visiting insect communities across the three habitats were different (Table 4).

When *q* = 0, the species abundances do not count at all, and *N*_0_ represents species richness. The species richness of plot 3 (15 spp.) and plot 2 (15 spp.) were much higher than that of plot 1 (12 spp.).

When *q* = 1, the number of species calculated by *N*_1_ is proportional to its abundance and can be interpreted as the effective number of common species in the community. The number of common insect species was about five species in plot 1, seven species in plot 2, and six species in plot 3.

When *q* = 2, which indicates the inverse Simpson index (*N*_2_), it can be interpreted as the effective number of dominant species in the community. Therefore, the number of dominant species in plot 2 was five, which were *P. zonata*, *A. cerana*, *Helicophagella melanura*, *Eristalis arvorum*, and *Vespa affinis*. The number of dominant species in plot 3 was four, which were *A. cerana*, *P. zonata*, *Musca domestica*, and *H. melanura*. The number of dominant species in plot 1 was 3, which were *P. zonata*, *H. melanura*, and *M. domestica*.

When *q* = 3, which indicates the inverse Berger–Parker index (*N*_3_) and can be interpreted as the effective number of the most dominant species in the community, it was significantly higher in plot 2 and plot 3 than in plot 1.

### 3.3. Pollination Behaviour of Flower-Visiting Insects and Selection of High-Potential Pollinators in Different C. oleifera Habitats

Further, we investigated the pollination behaviour of flower-visiting insects (Figure 6). When bees visited the flowers, they landed directly on the stamen cluster and crawled back and forth using their proboscis to suck nectar or inserted their heads into the base of the filaments beneath the stamens to feed. During this process, the head, thorax, abdomen, and legs came in direct contact with the stamens and stigma. Wasps primarily depleted nectar from *C. oleifera* flowers and occasionally foraged for pollen. They often drove away *A. cerana. Vespa bicolor* and *V. affinis* landed directly on stamens, inserting their heads into the base of the pistil to remove nectar, during which pollen on their bodies may contact the stigma and facilitate pollination. Flies (Syrphidae), on the other hand, typically perched directly on the stamens, moving around in search of pollen and seldom feeding on nectar. Among these, *P. zonata* has a body covered with dense setae that easily adhere to pollen, making it capable of pollination. Other flies, such as Calliphoridae, Sarcophagidae, Muscidae, and Gasterophilidae, mostly remained on the petals, rarely coming into contact with the stigma or crawling on wilted flowers. Their flower-visiting behaviour was minimal, and they generally lacked pollination capabilities.

Therefore, we combined the pollination behaviours of different dominant insect species and flower-visiting insects and found that the high-potential pollinators in plot 1 was *P. zonata*. In plot 2, *P. zonata*, *A. Cerana*, *E. arvorum*, and *V. affinis* were the high-potential pollinators. In plot 3, *A. Cerana* and *P. zonata* were the high-potential pollinators.

### 3.4. Individual Characteristics and Foraging Behaviour of the Main High-Potential Pollinators

A comparative analysis of the two high-potential pollinators revealed the following (Table 5): *P. zonata* had the larger body size, with a length reaching approximately 13 mm. There were significant differences in the pollen-carrying capacities of the two species. Despite its smaller size, *A. cerana* carried more pollen, averaging 15,383 grains per individual, while *P. zonata* had each individual carrying over 4367 grains.

In terms of foraging behaviour (Table 5), *A. cerana* was highly enthusiastic about collecting pollen and nectar. It frequently moved back and forth across the upper part of the anthers, using its middle and hind legs to rapidly brush and collect pollen. This behaviour typically lasted 2–10 s, or it probed the middle of the anthers to extract nectar for approximately 1 min. *A. cerana* could visit 3 to 15 flowers on the same plant before moving to a neighbouring plant, with a foraging frequency generally reaching 231.6 visits per hour and an average single-flower stay time of 14 s. *P. zonata* primarily foraged for pollen, typically remaining on the anthers or crawling around the reproductive parts of the flowers. It had the longest average single-flower stay time, ranging from 1 to 132 s, with an average of 17.36 s, resulting in the lowest foraging frequency of the three, at only 152.4 visits per hour.

### 3.5. Relationship Between Pollinator Abundance and Floral Characteristics

This study used floral traits as explanatory variables and insect abundance features as response variables to analyse the relationship between floral traits of different *C. oleifera* varieties and the abundance traits of three insect species. To reduce the interference of multicollinearity on the redundancy analysis (RDA) results, the variance inflation factor (VIF) was first calculated for all candidate explanatory variables, and variables were screened using a threshold of VIF < 10. Seven variables (stamen length, style length, manninotriose, fructose, sucrose, galactose, and stachyose) were ultimately selected, indicating no significant multicollinearity issues among these variables, making them suitable for inclusion in the subsequent RDA model.

Based on the screened variables, RDA was performed (based on correlation), and the results showed (Figure 7): RDA1 axis explained 29.43% of the variation in species distribution, serving as the primary ordination axis; RDA2 axis explained 4.71% of the variation, serving as the secondary ordination axis; the cumulative explanation rate of the two axes was 34.14% (*R*^2^ = 0.3414), indicating that the selected explanatory variables significantly explained the distribution patterns of insect abundance.

In terms of variable contribution and F-test values (Table 6), stamen length had the highest contribution (41.74%, *F* = 2.58), exerting the most significant influence on insect abundance distribution; style length had the second-highest contribution (21.89%, *F* = 1.27), jointly dominating the positive distribution of the RDA1 axis with stamen length.

The RDA ordination plot further revealed the association patterns between explanatory variables and insect abundance distribution: The arrows for stamen length extended the longest along the RDA1 axis, indicating a strong positive correlation between these variables and insect abundance (e.g., number of insect population and number of Diptera insects). In *C. oleifera*, flowers with highly exposed stamens are more easily detected and visited by pollinators, thereby enhancing pollination efficiency.

## 4. Discussion

Generally, a co-evolutionary relationship exists between insect-pollinated plants and their pollinators [25]. *C. oleifera* produces abundant nectar, which accumulates to significant levels by the third day of flowering, attracting pollinators to fulfil their pollination needs [26]. The first four days of flowering are optimal for pollination, as pollen viability and stigma receptivity peak during this period, coinciding with maximum nectar accumulation and pollinator visitation rates [27]. Further investigation into the relationship between floral characteristics and pollinators revealed that stamen length influences pollinator abundance. Flowers with highly exposed stamens are more easily detected and visited by pollinators, thereby enhancing pollination efficiency [28]. The high content of oligosaccharides, such as those found in cotton honeydew and sugar beets, in *C. oleifera* nectar may reduce the populations of *A. mellifera ligustica* and *A. cerana*, which are susceptible to poisoning [17]. Other pollinators, such as hoverflies (Syrphidae), are generalist pollinators capable of visiting a wide range of plant species. They play a critical role in environments with low pollinator diversity and have become the main pollination vectors, partially compensating for the shortage of bees [16]. Given the global crisis of declining honeybee populations [29,30], the development of effective antitoxins and the implementation of sustainable pollination strategies are essential for *C. oleifera* cultivation [31].

In this study, 22 species of pollinating insects belonging to 13 genera in 8 families of the orders Hymenoptera and Diptera were identified across 3 different habitats. Hill numbers were used to quantify biodiversity across the plots [22]. The type and abundance of dominant pollinators varied among habitats: plot 1 exhibited lower species richness, rarity (i.e., prevalence of low-abundance species), and dominance compared to plots 2 and 3. Higher pollinator diversity mitigates the impacts of environmental changes, such as extreme climate events and habitat destruction, which can severely reduce pollinator community diversity. When dominant pollinators decline, diverse communities maintain ecosystem stability via functional redundancy and compensatory effects from functionally similar species [32]. For example, climate change, vegetation shifts, and land-use intensification directly reduce wild bee populations [33,34]. In plot 1, near-absence of wild bees drastically limited pollination efficiency; however, hoverflies compensated by acting as primary pollinators. The asynchronous responses of different pollinator taxa to disturbances (e.g., temperature fluctuations and habitat fragmentation) further buffer pollination networks against collapse by preserving key functional groups. Additionally, soil compaction in intensively managed forests disrupts pollinator nesting and reproduction [35,36]. All three plots in this study were fully reclaimed and intensively managed, leading to the removal of native ground cover plants, reduced nectar plant diversity, and subsequent declines in wild bee populations. For instance, the low abundance of Apidae and high dominance of *P. zonata* in plot 1 likely resulted from its large size (340 ha) and complete land clearing, which destroyed bee habitats. In contrast, plots 2 and 3 (20–30 ha) were surrounded by forests, preserving habitat connectivity and preventing population collapse [37]. Thus, retaining patches of forest vegetation at mountaintops or foothills during large-scale *C. oleifera* cultivation is recommended to sustain wild bee habitats and biodiversity [38].

High-potential pollinators varied by habitat, with *A. cerana* and *P. zonata* being the most frequent in *C. oleifera* orchards. Analysis of their morphology and foraging behaviour revealed that pollen load differences depended on body size, hair density, and hair structure [39,40]. *A. cerana* carries significantly more pollen due to its dense body hair [15], and its high visitation frequency makes it the most high-potential pollinators. However, *P. zonata* compensates through high population density and tolerance to nectar toxins [41]. Hoverflies also exhibit greater activity under low temperatures and higher resilience to pesticides and habitat disturbance compared to Hymenoptera [42]. Increased floral visitation rates generally correlate with higher pollinator abundance and foraging frequency, thereby enhancing pollination success [43]. Artificial introduction of managed pollinators (e.g., honeybees) has proven effective in improving crop yields [44,45,46], suggesting that protecting pollinator resources, maintaining floral diversity, and creating bee-friendly habitats are critical.

## Figures and Tables

**Figure 1 insects-16-00282-f001:**
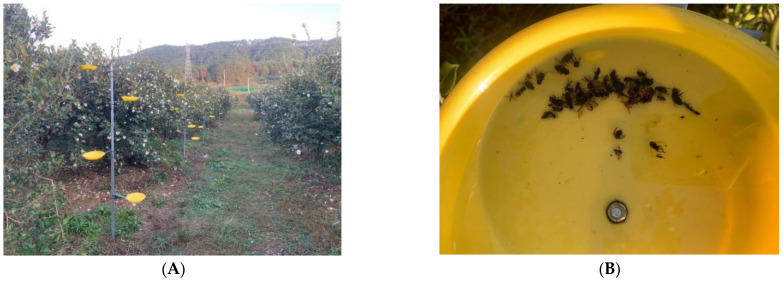
Investigation of pollinating insects in *C. oleifera* plots (trap device method). (**A**) Trapping devices; (**B**) Yellow-trapping plate.

**Figure 2 insects-16-00282-f002:**
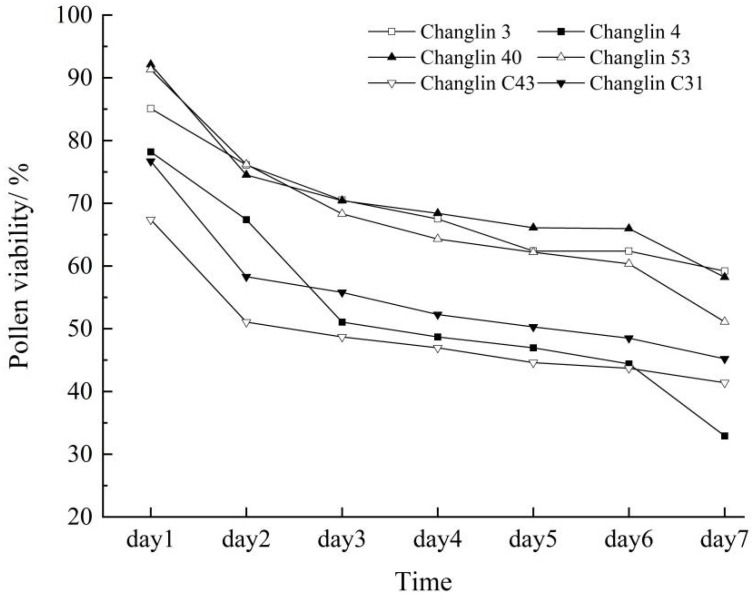
Pollen viability of different *C. oleifera* varieties.

**Figure 3 insects-16-00282-f003:**
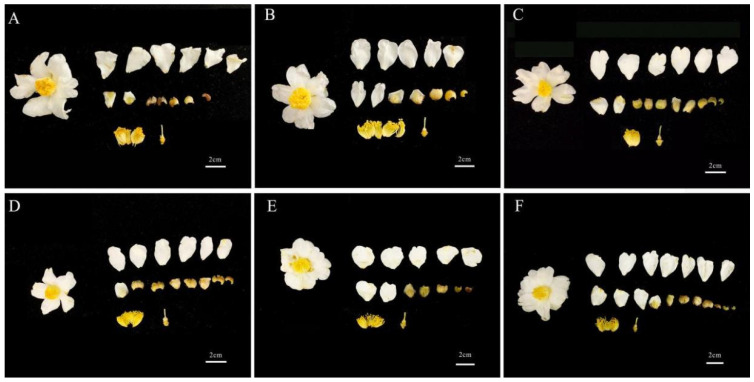
Images of floral morphology from different *C. oleifera* varieties. (**A**) Changlin 3; (**B**) Changlin 4; (**C**) Changlin 40; (**D**) Changlin 53; (**E**) Changlin C31; (**F**) Changlin C43.

**Figure 4 insects-16-00282-f004:**
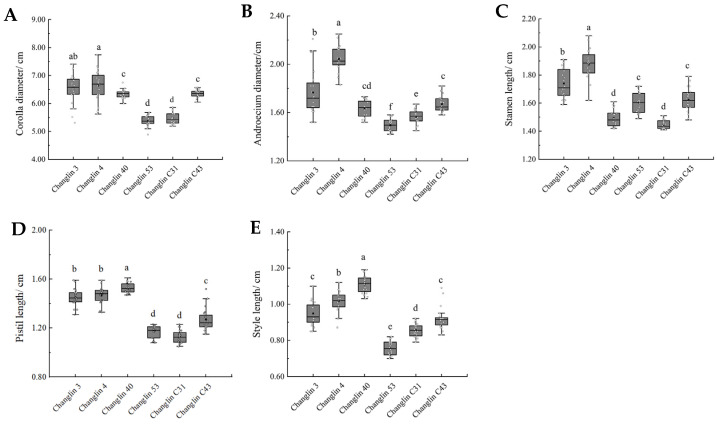
Floral trait characteristics of different *C. oleifera* varieties. (**A**) Corolla diameter; (**B**) Androecium diameter; (**C**) Stamen length; (**D**) Pistil length; (**E**) Style length. Different letters indicate significant differences at *p* < 0.05.

**Figure 5 insects-16-00282-f005:**
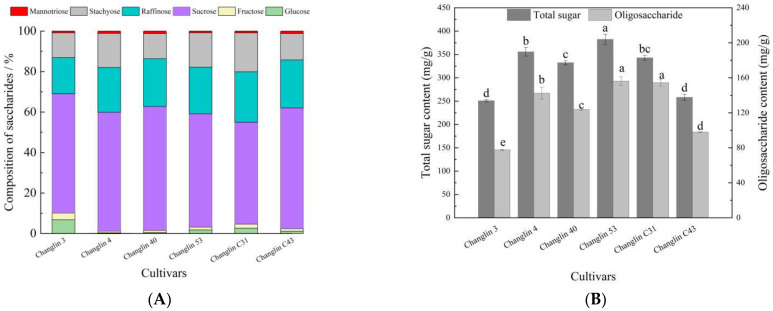
Sugar composition content in nectar of different *C. oleifera* varieties. (**A**) Composition of saccharides in nectar of different *C. oleifera* varieties; (**B**) Total sugar and oligosaccharides in nectar of different *C. oleifera* varieties. Different letters indicate significant differences at *p* < 0.05.

**Figure 6 insects-16-00282-f006:**
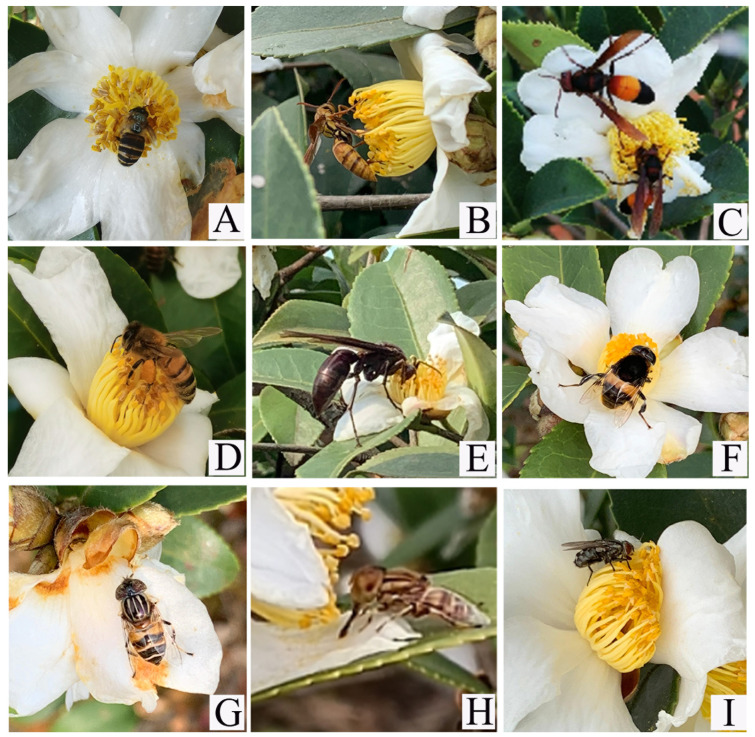
Insect flower visitors for *C. oleifera*. (**A**) *Apis cerana*; (**B**) *Vespa bicolor*; (**C**) *Vespa affinis*; (**D**) *Apis mellifera ligustica*; (**E**) *Polistes gigas*; (**F**) *Phytomia zonata*; (**G**) *Eristalis arvorum*; (**H**) *Lathyrophthalmus arvorum*; (**I**) *Helicophagella melanura*.

**Figure 7 insects-16-00282-f007:**
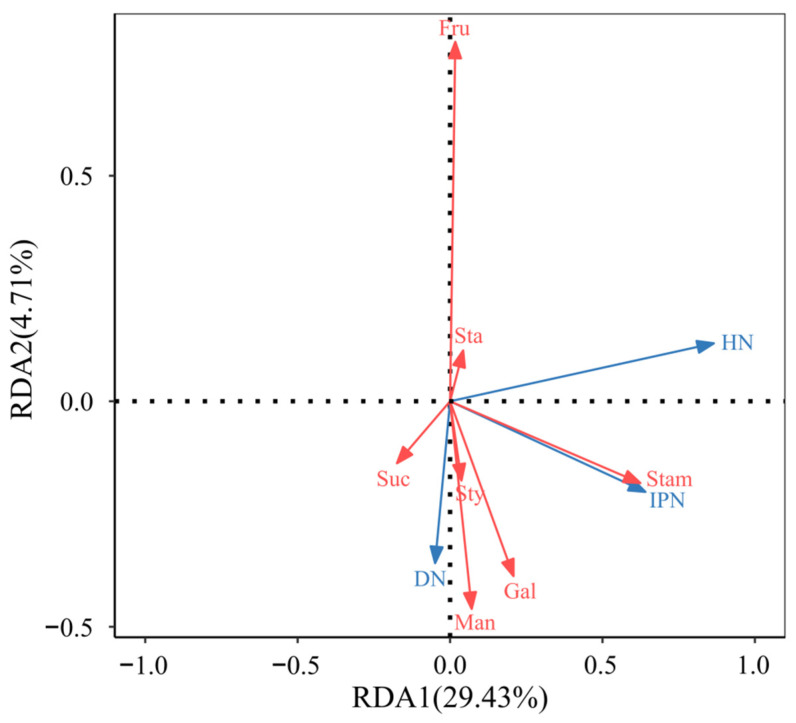
RDA biplot of *C. oleifera* floral traits and insect abundance. Stam: Stamen length; Sty: Style length; Fru: Fructose; Gal: Galactose; Suc: Sucrose; Man: Manninotriose; Sta: Stachyose; DN: number of Diptera insects; HN: number of hymenopteran insects; IPN: number of insect population.

**Table 1 insects-16-00282-t001:** Basic situation of different *C. oleifera* plots.

Experimental Site	Altitude (m)	Average Temperature (°C)	Relative Humidity (%)	Precipitation (mm)	Agrotype	Area (hm^2^)	Main Planting Variety
Youcheng Town	58	17.2	76.2	35.7	Yellow Soil	340	Changlin 4, 40, 3, 53
Qianshan Town	81	18.0	72.5	32.4	Red-Yellow Soil	22	Changlin 4, 40, 3, 53, C31, C43
Fengtian Town	96	18.8	73.4	55.6	Red Soil	30	Changlin 4, 40, 3, 53, C31, C43

Meteorological data were averaged from October to November.

**Table 2 insects-16-00282-t002:** Stigma receptivity of different *C. oleifera* varieties.

Time	Changlin 3	Changlin 4	Changlin 40	Changlin 53	Changlin C31	Changlin C43
day1	+ + +	+ + +	+ +	+ + +	+ +	+ +
day2	+ + +	+ + +	+ + +	+ + +	+ + +	+ + +
day3	+ + +	+ +	+ + +	+ + +	+ + +	+ + +
day4	+ +	+ +	+ +	+ +	+ +	+ +
day5	+	+/−	+/−	+ +	+/−	+ +
day6	+/−	−	−	+/−	−	+/−
day7	−	−	−	−	−	−

(−) stigmas with no receptivity; (+/−) part of the stigmas with receptivity; (+) stigmas with receptivity; (+ +) stigmas with higher receptivity; (+ + +) stigmas with the highest receptivity.

**Table 3 insects-16-00282-t003:** Trap-caught insects in *C. oleifera* populations at three experimental sites.

Family	Species	Proportion		
		Plot 1	Plot 2	Plot 3
Apidae	*Apis cerana*	2.91%	20.79%	37.31%
	*Apis mellifera ligustica*	0.87%	—	3.72%
Vespidae	*Vespa bicolor*	6.98%	3.56%	4.49%
	*Vespa affinis*	4.94%	6.14%	—
	*Vespa mandarinia*	2.03%	4.75%	—
	*Vespa ducalis*	—	—	1.08%
	*Vespa nigrithorax*			0.62%
	*Vespula flaviceps*	—	0.40%	0.62%
	*Polistes yokohamae*	—	0.40%	0.46%
	*Polistes gigas*	0.29%	0.20%	—
Scoliidae	*Campsomeris annulata*	0.87%	—	—
Syrphidae	*Phytomia zonata*	55.52%	32.48%	20.90%
	*Eristalis arvorum*	—	11.88%	—
	*Eristalis tenax*	—	—	0.31%
	*Lathyrophthalmus arvorum*	—	0.40%	0.31%
	*Episyrphus balteatus*	—	—	0.46%
Calliphoridae	*Stomorhina obsoleta*	0.87%	—	
	*Lucilia sericata*	5.23%	1.58%	2.01%
	*Chrysomya megacephala*	—	0.99%	8.36%
Sarcophagidae	*Helicophagella melanura*	9.88%	12.28%	8.98%
Muscidae	*Musca domestica*	9.59%	3.96%	10.37%
Gastrophilidae	*Tabanus* spp.	—	0.20%	—

**Table 4 insects-16-00282-t004:** Diversity of major flower-visiting insect communities in different sites.

	Species Richness (*N*_0_)	Exponential Shannon Index (*N*_1_)	Inverse Simpson Index (*N*_2_)	Inverse Berger–Parker Index (*N*_3_)
Plot 1	12	4.93 ± 0.98 b	3.07 ± 0.81 b	1.85 ± 0.31 b
Plot 2	15	7.02 ± 0.04 a	5.29 ± 0.09 a	3.08 ± 0.11 a
Plot 3	15	6.56 ± 0.38 a	4.69 ± 0.42 a	2.69 ± 0.24 a

Within columns, different letters indicate significant differences at *p* < 0.05.

**Table 5 insects-16-00282-t005:** Comparison of individual characteristics and foraging behaviour of dominant pollinators.

Species	Body Length/mm	Body Width/mm	Pollen Load (Grain/Individual)	Visiting Frequency/(Times/h)	Time Visiting Each Flower/s		
					Shortest	Longest	Average
*Apis cerana*	11.93 ± 0.77 b	4.39 ± 0.06 b	15,383 ± 3996 a	231.60 ± 68.83 a	2	76	13.98 ± 14.09 a
*Phytomia zonata*	13.14 ± 0.47 a	7.32 ± 0.25 a	4367 ± 1335 b	152.40 ± 40.81 b	1	132	17.36 ± 21.39 a

Within columns, different letters indicate significant differences at *p* < 0.05.

**Table 6 insects-16-00282-t006:** Contribution rate of floral characteristics to insect abundance features.

Factor	Contribution/%	*F*
Stamen length	41.74	2.58
Style length	21.89	1.27
Manninotriose	15.39	0.88
Fructose	10.11	0.61
Sucrose	5.06	0.29
Galactose	3.58	0.19
Stachyose	2.23	0.11

## Data Availability

The data that supports the findings of this study are available on request from the author.

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
