# Peer review of "Ecological Interactions Between Camellia oleifera and Insect Pollinators Across Heterogeneous Habitats"

_insects, 2025, doi:10.3390/insects16030282_

Round 1

Reviewer 1 Report (Previous Reviewer 3)

Comments and Suggestions for Authors

The manuscript is well-structured and shows substantial improvements compared to the first version. It is scientifically sound and makes a valuable contribution to the field of pollination ecology. The methodology is robust, the analysis is thorough, and the findings have practical implications for both agriculture and conservation.

However, there are several areas where the presentation, flow, and interpretation of results could be further enhanced.

In the results, we suggest adding a brief discussion explaining how floral traits directly influence pollinator behavior, as well as including a figure or graph summarizing species richness and diversity indices for better visualization. It would also be helpful to discuss the ecological implications of Diptera dominance in certain plots compared to Hymenoptera.

In the discussion, we recommend elaborating on broader ecological implications—such as how managing pollinator diversity can buffer against environmental changes—and comparing your findings more extensively with previous studies to highlight both similarities and differences.

Once these minor revisions are made to improve clarity, depth, and presentation, the manuscript will be suitable for publication.

Author Response

Thank you very much for taking the time to review this manuscript. Those comments are very helpful for revising and improving our paper, as well as the important guiding significance to our research. We have studied the comments carefully and have made a correction which we hope meet with approval. The following replies are the main comments made to you, and we would like to express our thanks to you once again.

Comments 1: In the results, we suggest adding a brief discussion explaining how floral traits directly influence pollinator behavior, as well as including a figure or graph summarizing species richness and diversity indices for better visualization. It would also be helpful to discuss the ecological implications of Diptera dominance in certain plots compared to Hymenoptera.

Response 1: Thank you for pointing this out. We have added a new paragraph in the Results section (lines 401–402) to explicitly discuss how specific floral traits (e.g.,  stamen length) influence pollinator behavior. For instance, flowers with highly exposed stamens are more easily detected and visited by pollinators.

Additionally, following the Academic Editor’s recommendation, we have incorporated Hill numbers (a unified diversity metric integrating species richness and evenness) into our analysis. This addition, now included in the Methods section (Lines 167–182) and presented in revised Table 4, allows for a more comprehensive interpretation of biodiversity patterns across the study sites (Lines 328–329).

Comments 2: In the discussion, we recommend elaborating on broader ecological implications—such as how managing pollinator diversity can buffer against environmental changes—and comparing your findings more extensively with previous studies to highlight both similarities and differences.

Response 2: Thank you for your suggestion. We have added a new subsection in the Discussion (lines 434–444) to explore the broader ecological implications of our findings. Specifically, we discuss how maintaining pollinator diversity—particularly the functional complementarity between Hymenoptera and Diptera—can enhance ecosystem resilience to environmental changes such as climate variability, habitat fragmentation, and pesticide exposure. For example, Diptera's tolerance to nectar toxins and their ability to thrive in disturbed habitats make them important backup pollinators in agricultural systems.

Reviewer 2 Report (New Reviewer)

Comments and Suggestions for Authors

Review report for Manuscript entitled “Adaptation of flower-visiting insects to the pollination biological characteristics of Camellia oleifera across heterogeneous habitats” Manuscript ID: insects-3421980 for potential article in insect

This study investigated the insect visitors to some varieties of Camellia oleifera and examined the relationship between the abundance of visitor species and floral traits that varied among varieties. The authors stated that this study was conducted to identify efficient or effective wild pollinator species for each variety of cultivated Camellia oleifera facing pollinator decline, but I'm afraid there remain several problems in this article as below. I advise the authors to completely reconstruct the MS to be faithful to the data they have and make it consistent throughout the introduction, method, and discussion. Since I found many problems in the methods, my comments are mainly focused on the method section.

Major comments:

First, the authors claim that they identified A. cerana and P. zonata as effective pollinators, but they did not investigate the pollen deposition and upcoming fertilization by these visitors. They say it only from the observation of visitation frequency and pollen load. So, strictly speaking, they are still "high potential" pollinators.

Second, although the authors seem to aim at reviewing the most efficient or abundant visitor species for each variety from the Introduction, they used only the abundance of pooled visitors (Hymenoptera, Diptera, or total visitors) as a response variable of RDA, which sounds nonsense.

Third, the poor explanation of sample size and replication of observations and statistical procedures (see below) makes it difficult to assess their reliability. In particular, it seems that the authors are not familiar with the methods for RDA and in some part mall application and misinterpretation may exist. I strongly recommend study before use.

Minor comments:

Abstract and Discussion

-Although the presence of this toxin in the nectar of Camellia oleifera is mentioned, it seems to come out of the blue. If the authors hope to discuss it in relation to visitor behavior, they should investigate it themselves or at least explain it in the introduction with references.

2. Materials and Methods

2.1 Study Area

-Since the study is intended to find the best wild pollinator for each variety in the context of wild pollinator decline caused by agricultural practices including pesticides, the authors should explain the management practices in the study sites. I think they should include several sites with less mowing and ideally no pesticides to find out the composition of the true pollinator communities and the best polliating species.

2.2 Experimental Methods

2.2.1 Floral Morphology and Biology

-Please explain the year and dates (at least the number of days would be required along with a range of periods, like 10 days during Oct 20 2020 to Nov 20 Nov 2020) for all the field survey.

-Please add references for the tests of pollen viability, stigma reseptivity and nectar sugar measurement method.

-The sample size for floral morphology is unclear. "20 samples per variety" means that the authors sampled only 6 or 7 replicates per site?

-The sample sizes for pollen viability and stigma receptivity tests and sugar content measurements are unclear. Please provide the exact numbers for each variety per site.

2.2.2 Floral Visitors and Potential Pollinators

2.2.2.1 Survey

-Please explain the year and dates (at least the number of days would be required along with a range of time periods, such as 10 days during Oct 20, 2020 to Nov 20, 2020) for the entire field survey.

-Were the 10 trees surveyed at each site the same throughout the experiment? I mean, did the authors use the same trees for camera trap, pan trap, observation, and netting (it seems they used different trees at least for netting. It's so confusing)? In addition, if these surveys were repeated several days, did the authors randomly select different trees or use the same trees throughout the survey days?

-Please clearly describe the composition of the solution used for trapping. What kind of chemicals did the authors use for the pan trap?

2.2.2.2 Visit behavior and body pollen load

-For insect sampling, how many observation units (10-minute observations) were made for each tree? Also explain the time and number of days. 

-or the measurement of single flower visit duration, how many replicates were made for each variety and for each insect taxonomic group? The authors mentioned replicates of 50, but was this in total or for what?

-I could not follow the calculation of visitation frequency. The authors write "VF = number of visits × number of flowers visited × number of plants visited / total observation time". I'm afraid that "number of visits" would be "number of individuals" and "number of flowers visited" could be "number of flowers visited per individual within trees" in this context. In addition, I do not understand why the authors included "number of plants visited", since there is no explanation for observation of movement among trees by an individual insect.

-For pollen load counting, the authors state that they pooled 2-3 individuals. Please explain that the authors converted the value to an individual basis.

2.3 Statistical Analysis

-Authors used one-way ANOVA for floral traits, but I'm afraid there might be differences between sites. Please consider.

-The explanation of the data for each statistical test is completely missing, especially for the visitor variables. I cannot see which data was used as the response variable for RDA. Also, the statistical test for RDA needs to be explained. It could be PERMANOVA.

-The statistical methods for the following comparisons are missing: comparisons of visitor composition and diversity indices between sites and cultivars, visitation frequency and duration of visiting a single flower between taxa.

-I could not understand how data from different observation methods (eye observation, trapping, and netting) were combined or used separately for analysis.

Results

-Actual numbers of visits or individuals must be reported in addition to proportions within sites.

-For RDA, authors should report the value of adjusted R^2 for the model. I think authors could reconsider the selection procedure of explanatory variables (floral traits). Authors seem to select two variables (androecium diameter and stamen length) for the final RDA just because the contribution of these factors is large, but these factors seemed to be highly correlated from Fig. 6. Strong multicollinearity should be avoided in RDA (Borcard et al. 2018. Numerical Ecology with R, Springer). The selection of explanatory variables should be based on the VIF value or, ideally, on a model selection procedure using AIC. Furthermore, the RDA plot should be drawn by the selected variables, but all measured variables are included in Fig. 6.

-In addition, the contributions of the two axes seem questionable because the total is 100%. In RDA of ecological data, the contribution of constrained factors cannot be 100%; the contribution of unconstrained factors always exists, in my understanding.

-The authors should explain whether the RDA plot is based on distance ( scaling 1) or correlation ( scaling 2).

Comments on the Quality of English Language

Recommend English editing. 

Author Response

Thank you very much for taking the time to review this manuscript. Those comments are very helpful for revising and improving our paper, as well as the important guiding significance to our research. We have studied the comments carefully and have made a correction which we hope meet with approval. The following replies are the main comments made to you, and we would like to express our thanks to you once again.

Reviewer 3 Report (New Reviewer)

Comments and Suggestions for Authors

The authors have examined the flowering traits and pollinator diversity across different habitats and found Apis cerana and Phytomia zonata as key pollinators. Androecium diameter and stamen length significantly influenced pollinator visits. The study provides a good base for pollinator conservation in the given region. My comments are:

Title: Revise the title: Ecological Interactions Between Camellia oleifera and Insect Pollinators across Heterogeneous Habitats

L25: Instead of flowering biological characteristics, why don’t you use floral traits or floral characteristics?

L28-29: Rewrite for clarity.

L30: Instead of flowering visiting insects, you can use insect pollinators.

L30 and 31: Orders are never italicized.

L32: Remove comprehensive. Comparison is enough.

L40-42: Replace: Insects play an important role in the reproduction of cross-pollinated plants.

L45: Replace: …pollinator service function of different crops, ultimately reducing their yield.

L63: Rewrite for clarity.

Overall: The introduction section needs complete revision. It is recommended to take help from a native English speaker for improving your expression and connection between sentences.

Table 1: Give names to the experimental site. Plot 1, 2, 3 does not look well.

Overall materials and methods: The authors have not mentioned the manufacturer details. For example, vernier caliper, pipette, tweezers, and so on. Also, the manufacturer names come with the name of the country and its state if any. Check with MDPI guidelines.

Figure 3A: Develop it as an independent figure. Divide each picture into A, B, C, D and so on.

Same comments for 3B.

Discussion: Now that you have obtained some good results, discuss only the top three to four findings. Rewrite the discussion so that each paragraph focuses on a single result. Again, a native English speaker or MDPI language editing can help you.

Comments on the Quality of English Language

The quality of English language expression requires significant revision.

Author Response

Thank you very much for taking the time to review this manuscript. Those comments are very helpful for revising and improving our paper, as well as the important guiding significance to our research. We have studied the comments carefully and have made a correction which we hope meet with approval. The following replies are the main comments made to you, and we would like to express our thanks to you once again.

Round 2

Reviewer 3 Report (New Reviewer)

Comments and Suggestions for Authors

The authors have addressed majority of comments.

Author Response

Thank you for your valuable feedback and acknowledge that our revisions address most of the comments. We appreciate the time and energy you devoted to improving the quality of the manuscript.

This manuscript is a resubmission of an earlier submission. The following is a list of the peer review reports and author responses from that submission.

Round 1

Reviewer 1 Report

Comments and Suggestions for Authors

Dear authors

In this study entitled “Adaptation of flower-visiting insects to the pollination biological characteristics of Camellia oleifera across heterogeneous habitats”, you investigated the floral biology, some floral traits of varieties (n = 6) of Camellia oleifera, in addition to the composition/diversity of the entomofauna and floral visitors/potential pollinators in experimental sites (n = 3) of the species located in three cities in China. The data will support the management and conservation of the species' pollinators and the creation of public policies in this regard. Therefore, this is a relevant study mainly because the species has great economic importance. Data collection required a lot of effort and dedication from the authors and appears to have been very careful.

However, there are several problems (highlighted in the text with comments and/or suggestions) that need to be resolved before the possible publication of the article. Some of them may originate from the translation into English. Therefore, I ask the authors to carefully analyze my recommendations, which aim to improve the article.

Thank you for the opportunity.

Best regards.

Comments on the Quality of English Language

Dear

English translation needs to improve.

I left some suggestions in the "Introduction" and "Materials and Methods".

Reviewer 2 Report

Comments and Suggestions for Authors

The authors are conducting research into the pollinators that visit camellia flowers and several characteristics of the flowers. The current decline in pollinators is thought to be causing serious problems in camellia production. For this reason, it is important to identify pollinators and make recommendations for maintaining pollination services in the future.
However, the results obtained are not very new.

It might be interesting if the paper discussed the differences between geographical pollinators and the differences in flower traits in each region, but at the moment I don't think it is worth publishing in the journal.

>>The potential pollinators have already been clarified in the previous study (13) cited by the authors. Therefore, the results of this study only add a little information about regional differences.

>>Although the flower traits are clarified, in the discussion, there are no recommendations for the conservation of pollinators based on these results. The content of L410-413 is not of a quality that can be called a recommendation, but is simply supplementary information. It is necessary to have a discussion based on the revealed flower resources and traits.

>>There are measurements of the shape and nectar, but why is there no information about the pollen? In L356, the author says that it is abundant pollen, but where is the evidence for this?

Comments on the Quality of English Language

Quality of English language is no problem.

Reviewer 3 Report

Comments and Suggestions for Authors

The present manuscript presents a comprehensive study on the relationships between floral traits and flower-visiting insects in different habitats for Camellia oleifera, an important oil-producing plant in southern China. The study is well structured and provides substantial data on pollinator diversity, insect visitation rates, and the key floral traits affecting pollination success. The findings have practical applications in improving pollination management for C. oleifera, which could enhance crop yield and sustainability.

However, there are some areas where improvements could enhance the manuscript's clarity, scientific robustness, and the impact of the study. Specifically:

The introduction provides a good background on the importance of pollinators for C. oleifera, but it could be strengthened by adding a few more references to recent literature on the global decline of pollinators and its potential impact on oil crops.

In the "Materials and Methods" section, the descriptions of the experimental sites could benefit from more details on surrounding vegetation and land use that may impact insect diversity. Moreover, while the statistical methods are appropriate, it would be helpful to provide a brief justification for why redundancy analysis (RDA) was used instead of other multivariate techniques. Including the assumptions tested for each statistical approach would enhance the credibility of the analyses.

In the "Results", the authors should consider summarizing key findings in a table for easier reference, particularly for species abundance across habitats. Moreover, in Figure 5 and Table 2, it would be beneficial to add significance levels to indicate where differences in insect communities between plots are statistically meaningful.

The discussion adequately links the findings to existing literature and contextualizes the importance of the results. However, the ecological implications of pollinator decline, especially in the context of monoculture systems, could be explored further. Additionally, the authors should elaborate on practical recommendations for managing pollinator habitats in C. oleifera plantations, such as creating pollinator-friendly environments to mitigate the effects of habitat destruction.

In Figure 7 (RDA biplot), it would be useful to annotate key floral traits and insect species more clearly to make the relationships more intuitive for readers.

In general, the language is clear, but there are a few grammatical errors and awkward phrasings throughout the manuscript. Proofreading by a native English speaker or using language editing software would improve the readability. The authors sometimes use overly complex sentences that can be broken down for better comprehension, particularly in the results and discussion sections.

 Specific Comments:

  • Lines 18-19: "Apis cerana, Phytomia zonata, and Vespa bicolor were selected as the dominant pollinating insects." - Consider explaining briefly why these species were selected as the dominant pollinators. Adding a sentence about their specific traits or abundance could help.
  • Line 28: "essentiallybasically" should be changed to "essentially" or "basically."
  • Lines 29-31: "There were 22 species of flower-visiting insects from 14 genera and 8 families..." - This is a significant result that could benefit from a brief table or figure reference for better understanding. Consider adding a summary table with key species.
  • Line 66: "Flowering of C. oleifera occurs from late autumn to winter..." - Specify the months for clarity, as this may vary by region.
  • Line 72: "Domesticated bees are reluctant to collect nectar and pollen from C. oleifera flowers." - Consider including a citation here to support this statement, as it is important for understanding the challenges faced in pollination.
  • Lines 127-138: The description of survey methods could be enhanced by including a reference to any previous studies that have successfully used these methods or by clarifying how the trapping device was an improvement over others.
  • Lines 249-257: Consider explaining why Plot 2 showed greater diversity, evenness, and richness compared to other plots. Providing some insight into the differences in habitat management or environmental factors could make the analysis more meaningful.
  • Lines 283-285: Add a brief explanation about the significance of selecting A. cerana, V. bicolor, and P. zonata as the dominant pollinators. How does their behavior or abundance contribute to the pollination process effectively?
  • Lines 333-341: The redundancy analysis (RDA) results are significant, but a clearer explanation of what the axes represent would help the reader understand the results better. Consider adding a brief sentence on the interpretation of RDA and what each floral trait's influence suggests about the insect populations.
  • Line 354: "Camellia oleifera produces abundant pollen resources..." - The phrase "pollen resources" could be more precise. Consider adding details about the types of pollen or specific characteristics that are beneficial for pollinators.
  • Lines 373-374: "Given the global crisis of declining honeybee populations..." - It would be beneficial to provide a recent reference that discusses the decline in honeybee populations to strengthen the argument.
  • Lines 403-405: "Apis cerana has dense body hairs that allow it to carry significantly more pollen than other insects." - Consider adding a reference to a study that quantifies this pollen-carrying capacity to support the claim.
  • Figure 4: The sugar composition graph is informative, but consider adding error bars to show the variability within the data.
Comments on the Quality of English Language

In general, the language is clear, but there are a few grammatical errors and awkward phrasings throughout the manuscript. Proofreading by a native English speaker or using language editing software would improve the readability. The authors sometimes use overly complex sentences that can be broken down for better comprehension, particularly in the results and discussion sections.
